# Reliability of IL-6 Alone and in Combination for Diagnosis of Late Onset Sepsis: A Systematic Review

**DOI:** 10.3390/children11040486

**Published:** 2024-04-18

**Authors:** Julia Eichberger, Elisabeth Resch, Bernhard Resch

**Affiliations:** 1Research Unit for Neonatal Infectious Diseases and Epidemiology, Medical University of Graz, Auenbruggerplatz 34/2, 8036 Graz, Austria; julia.eichberger@medunigraz.at (J.E.); elisabeth.resch@medunigraz.at (E.R.); 2Division of Neonatology, Department of Pediatrics and Adolescent Medicine, Medical University of Graz, Auenbruggerplatz 34/2, 8036 Graz, Austria

**Keywords:** Interleukin-6, late onset sepsis, diagnostic accuracy, sensitivity and specificity, meta-analysis

## Abstract

Diagnosis of neonatal sepsis is difficult due to nonspecific signs and symptoms. Interleukin-6 (IL-6) is a promising marker for neonatal sepsis. We aimed to test the accuracy of IL-6 in neonates after 72 h of life in case of late onset sepsis (LOS). We searched for studies regarding IL-6 accuracy for the diagnosis of LOS between 1990 and 2020 using the PubMed database. Following study selection, the reported IL-6 sensitivities and specificities ranged between 68% and 100% and 28% and 100%, with median values of 85.7% and 82% and pooled values of 88% and 78% (respectively) in the 15 studies including 1306 infants. Subgroup analysis revealed a better sensitivity (87% vs. 82%), but not specificity (both 86%), in preterm infants compared to term infants or mixed populations. Early sample collection revealed the highest sensitivity (84%), but had the lowest specificity (86%). To assess quality, we used a STARD checklist adapted for septic neonates and the QUADAS criteria. Limitations of this review include the heterogeneous group of studies on the one side and the small number of studies on the other side that analyzed different combinations of biomarkers. We concluded that IL-6 demonstrated good performance especially in the preterm infant population and the best results were achieved by measurements at the time of LOS suspicion.

## 1. Introduction

The definition of late onset sepsis (LOS) includes presentation after the first 72 h of life and association with the postnatal nosocomial or community environment [1]. Neonates in the NICU are prone to LOS due to their immaturity and their lack of maternal protection by maternal antibody transfer in the case of very preterm infants [2]. Coagulase-negative staphylococci (Gram-positive cocci) represent the most common organisms causing nosocomial infections followed by Gram-negative bacilli and fungi [1,3]. Risk factors for the development of LOS besides immaturity are mechanical ventilation, intravascular catheterization, formula feeding, prolonged duration of intravascular access devices in cases of parenteral nutrition, any surgery, underlying respiratory and cardiovascular disease, and prolonged hospitalization [4]. In high-income countries, the mortality rate due to neonatal sepsis (including both early and late onset sepsis) ranges from 5% to 20% and higher mortality rates of over 70% can be observed in low- and middle-income countries (LMICs) [4]. Early and efficient treatment reduces both mortality and morbidity in neonates with suspected sepsis [5]. Hence, there is a great need for biological markers that immediately increase in cases of inflammation [6].

Released within 2 h after the onset of bacteremia, the levels of pro-inflammatory cytokine Interleukin-6 (IL-6) increase earlier than both PCT and CRP in neonatal septic patients [7,8]. IL-6 levels have been shown to be significantly elevated up to 48 h prior to clinical signs of sepsis [9]. Measured at the time of sepsis suspicion, IL-6 levels were found to be associated with sepsis severity and mortality risk in preterm infants [7]. Combinations of IL-6 with later and more specific biomarkers (e.g., CRP) have been reported [10]. 

The aim of this systematic review was to determine the accuracy of IL-6, both alone and combined with other markers, for the diagnosis of LOS by reviewing studies published between 1990 and 2020 and to explore the affecting factors. In this meta-analysis, we decided to focus solely on LOS due to the fact that the type of sepsis had previously been recognized as a source of heterogeneity [11].

## 2. Material and Methods

We used the Pubmed database to search for diagnostic accuracy studies of IL-6 in neonates published between 1990 and 2020 that proved the diagnostic capacity of IL-6. The search terms we used in combination were the following: (Interleukin-6 OR IL-6) AND (neonatal sepsis OR neonatal infection OR sepsis) AND (late onset sepsis OR LOS OR LONS). We did not need any PubMed filters or language restrictions.

We identified potentially suitable studies by screening the headlines of the studies and the abstracts. The following criteria had to be fulfilled by reviewing the abstract: only neonates presenting with culture proven and/or clinically suspected sepsis and IL-6 (alone or combined with other inflammatory markers) being evaluated regarding its potential for the diagnosis of LOS. We excluded all studies dealing with early-onset sepsis or other bacterial infections, all studies written in other languages than English or German, animal and in vitro studies. In line with the PRISMA criteria (see Figure 1), full text articles were screened for other potentially relevant studies. The following data were extracted from all full-text studies included in the analysis: first author, country, year of publication, definition of LOS, number of neonates, recruitment characteristics, reference standards, analysis of blood samples, and time of sample collection. Finally, the IL-6 test method and its use alone or combined with other markers were documented. All analyses were based on already published studies; thus, no informed consent or approval of the local ethic committee were required. Two investigators (JE, ER) independently performed the data extraction of all the included studies. In the case of discrepancies or disagreements during data extraction, the third reviewer (BR) resolved any differences.

We used an adapted STARD checklist for septic neonates as published by Chiesa et al. to assess the study quality [12]. This checklist included 25 items from the key domains; descriptions of participant recruitment, reference standards and index tests, which are answered with either yes or no [12]. We additionally performed a quality assessment of the diagnostic accuracy studies (QUADAS) tool including 11 questions. Questions with “yes”, “no”, and “unknown” answers were scored as 1, −1, and 0, respectively [13]. Thus, we could confirm our first analysis using the STARD criteria.

We explored causes for heterogeneity by means of subgroup analysis. The influence of gestational age was evaluated by comparing subgroups of preterm and mixed populations. The timings of sample collection and its influence on IL-6 accuracy were analyzed by dividing the studies into those reporting sample collection at the time when sepsis was suspicious, and those reporting collection times earlier than 12 h, earlier than 24 h and earlier than 48 h after suspicion of sepsis. Blinded studies and those with blood-culture-proven sepsis both formed individual subgroups. Biomarker combinations were assessed if at least three studies were found. For the subgroup analysis, we calculated the 2-by-2 tables of true positives, true negatives, false positives, and false negatives of each study from the data provided by the individual studies. We then pooled these values according to the subgroup. Calculating the sensitivity and specificity based on these values gave us the pooled sensitivity and specificity for this subgroup, or the overall sensitivity and specificity in the case of pooling all studies.

Finally, to summarize the results of the primary studies, a summarized ROC (sROC) curve was generated using the Moses–Littenberg method [14,15]. The homogenous area under the curve (AUC) was calculated using the formula provided by Rosman et al. [14].

## 3. Results

Figure 1 depicts the search strategy that finally identified 107 records, and 2 further articles were found in the reference list from the selected studies. After the exclusion of studies based on their titles, 47 abstracts were screened and 23 full text articles assessed. Finally, 16 studies [2,3,9,10,16,17,18,19,20,21,22,23,24,25,26,27] were eligible for meta-analysis. One study [25] only analyzed the biomarker combination IL-6 and CRP, leaving fifteen studies including a total of 1306 infants for the subgroup analysis of IL-6 as a single marker.

Almost all studies (n = 13) defined LOS as sepsis occurring after the first 72 h of life [2,10,16,17,18,19,20,21,22,23,24,25,26]. Two studies did not provide a definition but all included infants were older than 3 days [3,27]. Only one study defined LOS as sepsis >48 h [9]. Eight studies included only preterm infants [9,10,16,22,23,24,27,28], while the other eight studies included a mixed study population [2,3,17,18,19,20,24,25]. All studies measured IL-6 levels in peripheral blood. The majority of studies included cases with proven and clinical sepsis [2,3,9,10,16,18,19,20,24,27]. In two studies, the sepsis group consisted only of culture-proven cases [23,24]. One study performed separate analyses for cases of culture positive and cases of clinical sepsis [17].

The IL-6 sensitivities and specificities ranged from 68% to 100% and 28% to 100%, respectively; and the median values were 85.7% and 82%, as shown in Figure 2. The pooled sensitivity was 88% (95% CI: 85–90%) and the pooled specificity was 78% (75–81%), as shown in Figure 3. We summarized all the data extracted from the selected studies in Table 1 for IL-6 as a single marker and all the data for IL-6 in combination with other biomarkers in Table 2. 

Subgroup analyses are shown in Table 3. The sensitivity was higher in the preterm population (87% vs. 82%), while specificity was the same for both study populations (86%). Eleven studies [3,16,17,18,19,20,21,22,23,24,26] collected blood samples at the time of sepsis suspicion (0 h). Three studies [17,21,27] collected their samples earlier than 12 h after initial sepsis suspicion, six studies [10,17,21,22,24,27] earlier than 24 h and three studies [10,17,24] earlier than 48 h. One study [9] collected samples within a certain time interval rather than at a specific time point, and thus could not be assigned to one of the subgroups. Collecting the sample at the time of sepsis suspicion showed the highest sensitivity (84%), but the lowest specificity (86%) when compared to the later collection times, as follows: sensitivities and specificities of 57% and 94% before 12 h, 54% and 88% before 24 h and 67% and 92% before 48 h. In three studies [17,23,24], the sepsis group was formed by culture-proven cases only and summarizing these studies resulted in a pooled sensitivity and specificity of 85% and 74%, respectively. This corresponds to a decrease of almost 10% in IL-6 specificity, when compared to subgroups with similar sensitivity. Three studies [22,24,25] reported that their researchers were blinded to the results of the index test and the reference standard, while one of these studies only analyzed biomarker combinations [25]. Hence, blinding only formed part of the study design in two studies eligible for subgroup analysis. With 81% and 80%, both sensitivity and specificity were lower than in the preterm groups, mixed study population and sample collection at the time of sepsis suspicion, as shown in Table 4.

Seven studies reported the results of biomarker combinations including IL-6 [3,10,16,18,21,22,25]. Four studies combined the early sepsis marker IL-6 with CRP [3,10,16,25]. Combinations with early markers sTREM-1 (soluble Triggering Receptor Expressed on Myeloid Cells-1 [18] and CD64 (Cluster of Differentiation 64, n = 1)) [21] were studied by one study each. Combinations of up to three biomarkers including, in addition to IL-6, the markers IP-10 (Interferon gamma-induced protein 10), IL-10 (Interleukin-10), CRP and TNF-α (Tumor necrosis factor-α) have been investigated by Ng et al. [10,22]. The positivity criterion of the test was defined by Ng et al. [10,21,22] as any one marker above the cut-off level and by Dillenseger et al. [25] as one of the two above the cut-off level and not specified in the remaining studies. In the four studies analyzing a combination of IL-6 and CRP at sepsis suspicion [3,10,16,25], cut-off values ranged from 21.7 to 60 pg/mL and 4.05 to 14 mg/L, respectively, sensitivities ranged between 78.12 and 100% and specificities between 41 and 96%. The biomarker combination of IL-6 and CRP, measured at the time of sepsis suspicion, had the highest overall sensitivity (92%), but the lowest overall specificity (79%) in the subgroup analysis. 

Table 4 summarizes the quality assessment of the studies according to the adapted STARD criteria. All 16 articles [2,3,9,10,16,17,18,19,20,21,22,23,24,25,26,27] were studies on the diagnostic accuracy of IL-6, and the majority came from single perinatal centers. In 12 studies (75%), the enrolment of patients was solely based on clinical signs suspicious for sepsis. In two studies, cases were already diagnosed or had been excluded. Twelve studies (75%) were found to have different reference standards for the diagnosis of LOS and for verification of index test results; thus, we documented verification bias [2,3,10,16,18,19,20,22,25,26,27,28]. Only four studies (25%) used a composite reference standard for exclusion of LOS [16,21,25,27]. In two studies, we found CRP being a comparator of the index test and being part of the reference standard [23,25]. Clinical and demographic data were reported in 15 (94%) studies [3,9,10,16,17,18,19,20,21,22,23,24,25,26,27]. Four studies (25%) reported the number of neonates fulfilling inclusion criteria that failed to undergo the index tests and/or the reference standard [9,16,24,26]. All studies defined their cut-off values post hoc. AUC values were only reported in 8 out of the 15 included studies. Fortunately, in five of these studies, the AUC was above 0.9, which corresponds to an excellent diagnostic test, and above 0.75 in another two studies, which still corresponds to a good biomarker. Only in one study and for one time point in another study was the AUC below 0.75 [29]. Three studies (19%) reported details of the persons who executed the data analysis (number, training and expertise); additionally, four studies provided blinding information [2,22,24,25]. Measures of statistical uncertainty were reported in six (38%) studies [3,16,17,18,25,27]. Five studies (31%) provided information on calculation methods for test reproducibility [3,9,10,16,21]. Two studies included a cross-tabulation of the results [13,26], and only one study reported the process of how analyses were performed in case of unclear results, absent responses or outliers of index tests [2]. None of the studies reported illness severity scores or their distribution in neonates with and without LOS.

Figure 4 depicts the sROC curve summarizing the results of individual studies. The overall AUC was 0.88, which corresponds to a good diagnostic test [29].

## 4. Discussion

Our systematic review revealed a satisfying pooled sensitivity of IL-6 as a single marker of 88% (95% CI: 85–90%), and a lower pooled specificity of 78% (75–81%). Another review that included 31 studies incorporating 1448 infants demonstrated a global sensitivity of 82% (77–86%) and specificity 88% (83–92%), respectively [30]. Only 9 out of the 31 studies (29%) [3,9,10,16,17,18,22,24] from this review [30] coincided with studies in our review. This fact was mainly due to the missing differentiation between early and late onset sepsis in their meta-analysis. Other differences compared to our meta-analysis were the selection process on how the studies were selected, missing differentiation by gestational age and time of sampling, as well as combinations of IL-6 with other markers.

Fifteen studies analyzed the diagnostic accuracy of IL-6 as a single marker. Most studies measured IL-6 levels at the time of first signs and symptoms of sepsis. Küster et al. [9] in turn investigated the time course of IL-6 expression and its prognostic power in sepsis diagnostics. IL-6 was found to be superior to CRP in the prediction of sepsis 1 or more days before clinical diagnosis. The sepsis-proven group showed a significant increase in IL-6 levels from median baseline values of 7.5 pg/mL to 89.7 pg/mL on day −2, i.e., 2 days before clinical diagnosis [9]. Multiple studies found that IL-6 was only able to differentiate between sepsis and no sepsis at the onset and had limited potential for diagnosis later during the course of sepsis [18,27]. This is logical due to the early eruption of IL-6 and its short half-life time. Lusyati et al. [17] made serial determinations of IL-6 levels (0, 4, 12, 24, and 48 h). Despite decreasing IL-6 values at all five time points, significantly higher values were found in the proven sepsis group than in the control group for all five measurement points [17]. In the study by Panero et al. [26], all 51 patient controls had IL-6 concentrations <15 pg/mL, while the 17 patients with LOS had IL-6 levels strikingly greater than 15 pg/mL at presentation, corresponding to a sensitivity and specificity of 100% for IL-6. Gonzales et al. [24] found that IL-6 had a sensitivity of 75%, specificity of 68%, an NPV of 87% and PPV of 50% on day 0 of the sepsis episode. On day 1, the specificity and NPV improved to 90% [24]. However, their cut-off value of 18 pg/mL was defined solely upon inspection of the data [24].

Seven studies included in the meta-analysis reported results of biomarker combinations including IL-6 [3,10,16,18,21,22,25]. Raynor et al. [2], analyzing seven cytokines, found IL-6 to be the best-performing individual cytokine. IL-6 at a cut-off of 130 pg/mL demonstrated 100% sensitivity and 52% PPV when discriminating between patients without sepsis and those with sepsis (clinical or culture proven) [2]. Testing all 127 possible cytokine combinations for ruling out sepsis revealed that adding any other cytokine to IL-6 did not result in a higher PPV [2]. Ng et al. [10] identified IL-6, TNF-α and CRP as the best three markers for LOS diagnosis. A comparison of the diagnostic value of the individual markers versus a combination or panel of markers revealed higher sensitivity and better negative predictive values for the latter [10]. Serial measurements of inflammatory markers can further improve diagnostic accuracy. The highest sensitivities (98%) and specificities (91%) were reached when CRP and IL-6 were measured at day 0 combined with either TNF-α (day 1) or CRP (day 2) [10]. In a later study, Ng et al. [19] combined IL-6 and CRP at day 0 with CD64 at 24 (day 1), which resulted in a sensitivity and specificity of 100% and 86%, respectively. Sarafidis et al. [18] found the diagnostic accuracy of IL-6 combined with sTREM-1 (sensitivity and specificity 90% and 62%, respectively) not superior to that of IL-6 alone (sensitivity 80% and specificity 81%). The combination of IL-6 and CRP at time point 0 was superior to other markers and possible combinations in a study by Dillenseger et al. [25]; however, a sensitivity of 78% and specificity of 76% were not sufficient. Comparing two cut-off points, IL-6 at 60 pg/mL was shown to have good specificity (96%), but low sensitivity (67%), while a lower cut-off of 30 pg/mL had excellent sensitivity (100%) but only average specificity (74%) [16]. Combining the sensitive IL-6 (cut-off of 30 pg/mL) with the more specific CRP, sensitivity and specificity for sepsis prediction improved to 100% and 96% [16]. Comparing the diagnostic potential of the three markers CD64, IL-6, and CRP in combinations versus individual markers revealed only marginal improvement of sensitivity and negative predictive values [21].

Subgroup analysis was used to analyze the influence of the gestational age and the time of sample collection. One study [2] modified their cut-off criteria in order to achieve a sensitivity of 100%. To prevent introducing bias, this study [2] was excluded from the subgroup analysis. Some groups provided multiple results, e.g., for varying cut-off levels. To avoid introducing the same study population multiple times when comparing preterm versus mixed study populations, each study was included only once. We chose analyses including the whole study population and in cases of different scenarios, we chose those that yielded the best results [31].

Chiesa et al. [12] analyzed IL-6 diagnostic accuracy studies and found that the majority were suboptimal due to missing information on essential parts like the study design, conduct, analysis and interpretation of test accuracy [31]. We used the adapted STARD checklist [12] to analyze the quality of the present studies. Twelve of the sixteen included studies used different reference standards for diagnosing LOS and verifying index test results [2,3,10,16,18,19,20,22,25,26,27,28]. The majority of studies included proven and clinical sepsis cases [2,3,9,10,16,17,18,19,20,25,27]. In two studies, the sepsis group consisted only of culture-proven cases [23,24]. None of the studies included illness severity scores in their study design. As an inflammatory marker, CRP serves as an important comparator of the index test; however, in two studies, it was also used for being the reference standard to diagnose sepsis [23,25]. All studies included defined cut-offs post hoc, with most of them using ROC analysis. In one study, the cut-off was chosen solely upon inspection of the data [24] and one study did not provide information on the origin of their cut-off value [26]. For further information on the importance of each item, we refer to a recent publication of our study group [31]. In brief, incorporation bias occurs if the index test or the comparator of the index test form part of the reference standard. The fact that the person interpreting the results of these tests would gain some knowledge of the results of the reference standard distorts the diagnostic ability of these tests. This holds true for markers related to, and biomarker combination including, the marker which forms part of the reference standard [31].

Regarding the clinical applicability of IL-6 for sepsis diagnosis, Dillenseger et al. [25] stated that cytokine assays require a minimum time of 85 min to obtain the results, which would be compatible with clinical decision making but nonetheless should be shortened. Compared to CRP, determination of cytokines is more elaborate and their assays are more expensive; therefore, many hospital laboratories are not able to perform these assays [3]. Most laboratories are not able to perform these expensive tests in test batteries that further hamper their clinical usefulness as early markers [21]. Others like Değirmencioğlu et al. [23] already implemented IL-6 into clinical routines. Raynor et al. [2] argue that it is unlikely to achieve a 100% diagnostic accuracy via cytokines, since a robust systemic inflammatory response might be absent in some cases of clinical or Gram-positive sepsis. Verboon et al. [3] measured IL-6 levels after 48 h of antibiotic treatment to find out whether IL-6 might support the decision about the duration of antibiotic treatment (7 to 14 days) in cases of confirmed bacterial sepsis and clinical recovery. They found that a rapid decrease in IL-6 at 48 h would justify the early discontinuation of antibiotics [3]. The findings of Ng et al. [10] led to the same conclusion for the serial measurement of IL-6 and CRP measured at the day of sepsis suspicion and CRP measured again two days later. While withholding antibiotic treatment at the onset of sepsis is not recommended, high sensitivity (98%) and negative predictive values (98%) of this combination indicate that antibiotics could be discontinued at 48 h if the infants were in good clinical condition [10]. This finding can only complement the already common practice of empirically treating the infant for at least 48 h while awaiting blood culture results. In the era of continuously monitored blood culture systems, several studies have even challenged this time frame [32]. A study investigating the time-to-positivity (TTP) of blood cultures in children with proven sepsis found that 90% of blood cultures were positive within 36 h, and in most cases even <24 h of incubation [32]. They concluded that discontinuing empirical treatment in the absence of a positive blood culture should already be considered after 24 and 36 h [32].

The strengths of the study can be outlined as follows: we eliminated the factor of uncertainty in many studies between early or late onset sepsis by including only cases of LOS. Subgroup analysis identified the type of sepsis as a significant source of heterogeneity [11,30]. The limitations of the study are that we investigated a heterogeneous number of studies in order to gain information (subgroup analyses) on IL-6 performance and possible influencing factors. This might have influenced the precision of the study negatively. It might be useful for future research to analyze individual factors causing heterogeneity within otherwise homogenous subgroups. Unfortunately, only a few studies looked at biomarker combinations.

Based on the findings of this review, IL-6 might be of use for the diagnosis of late onset sepsis in populations of preterm infants when measured at the time of sepsis suspicion. Evaluation of these results in the context of existing literature was difficult since other reviews on this topic included either mixed study populations or even sepsis groups consisting of early and late onset sepsis cases. To confirm the use of IL-6 in the diagnosis of LOS, further prospective studies on well-defined study populations and with well-defined sepsis criteria are needed.

## Figures and Tables

**Figure 1 children-11-00486-f001:**
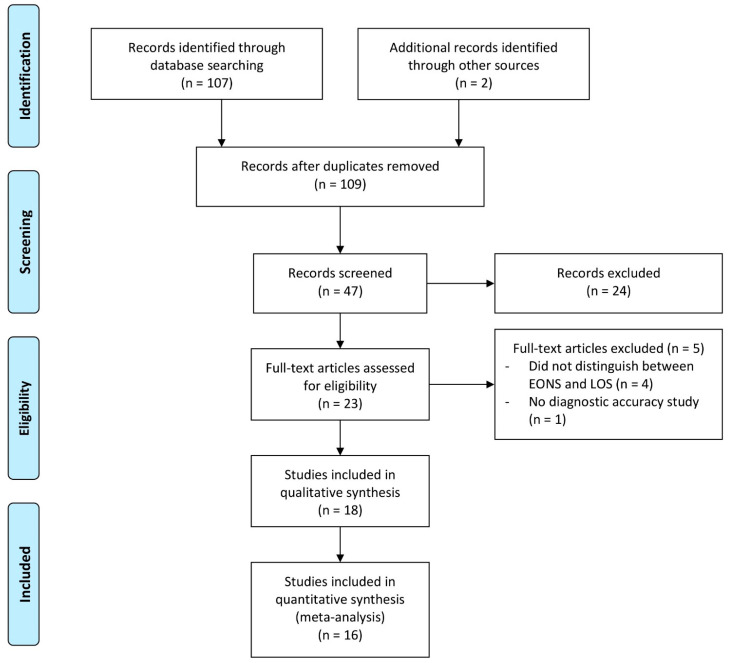
Flow chart of the study selection process for diagnostic accuracy of Interleukin-6 in late onset sepsis between 1990 and 2020. Reasons for the exclusion of 24 papers at abstract level were no diagnostic accuracy study (n = 9), exposom study (n = 1), biomarkers other than inflammatory markers (n = 3), language other than English (n = 2), animal study (n = 4), did not study or report outcomes of interest (n = 1), in vitro study (n = 3); dealt with EONS (n = 1).

**Figure 2 children-11-00486-f002:**
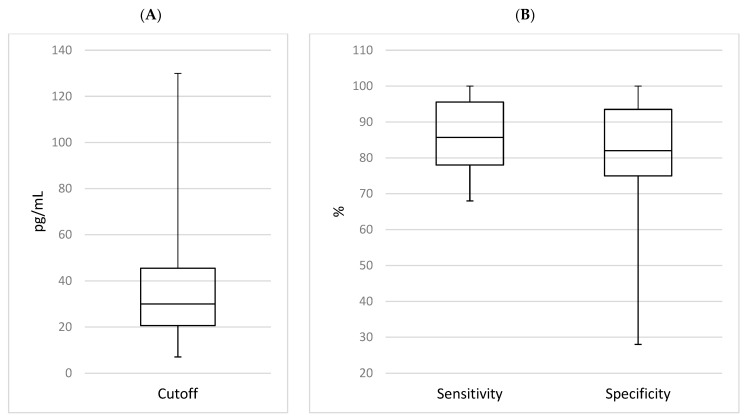
Boxplots of the distribution of IL-6 cutoff (**A**) and sensitivity and specificity values (**B**) of all diagnostic accuracy studies on late onset sepsis using IL-6 as a single marker.

**Figure 3 children-11-00486-f003:**
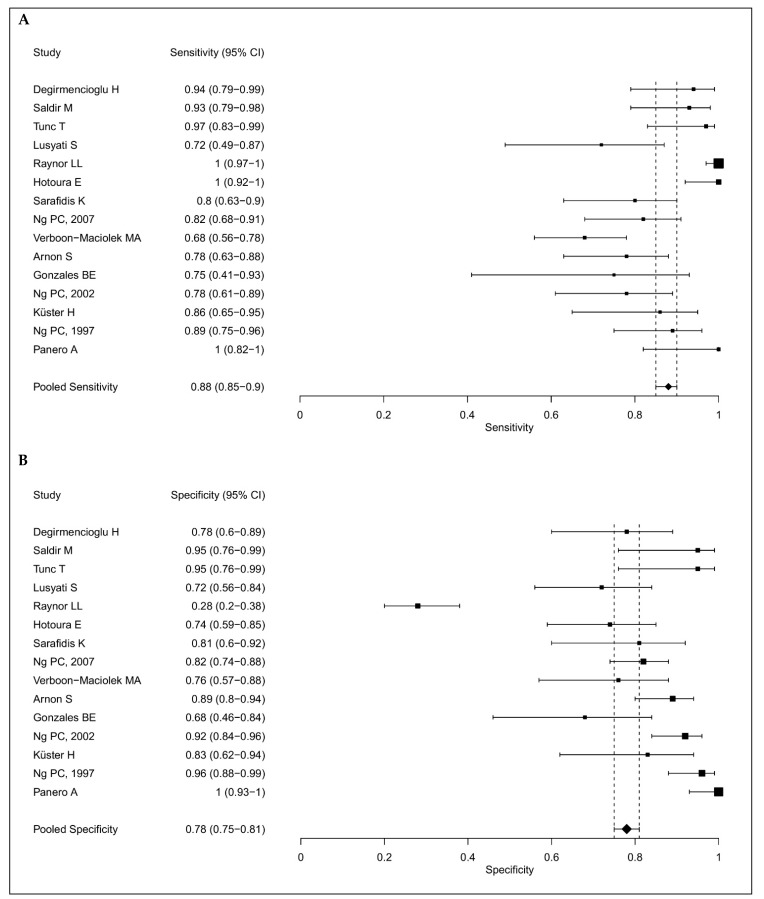
Forest plots showing the individual and pooled sensitivities (**A**) and specificities (**B**) of IL-6 diagnostic accuracy studies for the diagnosis of late onset sepsis [1,2,9,10,16,17,18,19,20,21,22,23,24,26,27].

**Figure 4 children-11-00486-f004:**
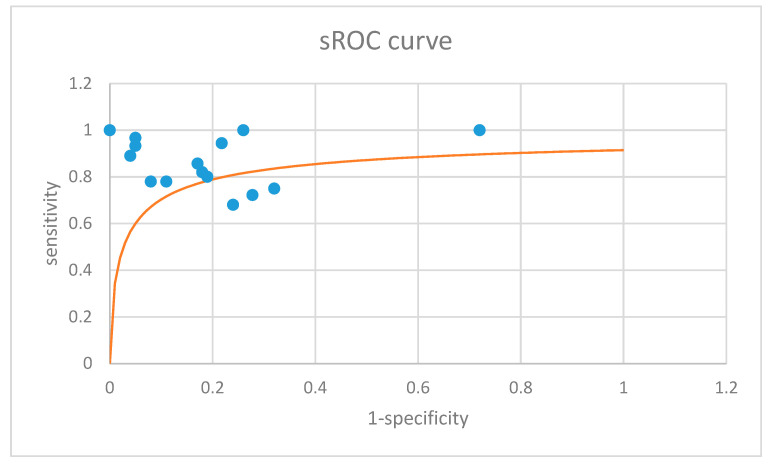
Summary receiver operating characteristics (sROC) curve (orange) summarizing the results of the 15 included primary studies (blue dots) [14,15]. The overall area under the curve (AUC) was 0.88 [14].

**Table 1 children-11-00486-t001:** Characteristics of IL-6 accuracy studies for the diagnosis of late onset sepsis using IL-6 as a single marker.

Author, Year, Country, Reference	LOS Definition	Recruitment	Reference Standard in Infected Neonates	Reference Standard in Control Neonates	Sample Studied, Time of Sample Collection	Test	IL-6 Cut-Off (pg/mL)	Sens, % (95% CI)	Spec, % (95% CI)	AUC (95% CI)	PPV, %	NPV, %
Değirmencioğlu H, 2019, Turkey [23]	>72 h	55 very preterm NICU infants (≤32 weeks): 26 infected (PS = 100%), 29 uninfected	Positive blood culture in addition to clinical signs and abnormal acute phase reactants	GA, birth-weight- and gender-matched infants with no signs or symptoms of sepsis	Neonatal serum, day 0 (after SS, at enrollment)	Solid phase, enzyme labeled, chemi-luminescent sequential immunometric assay	23.22 (ROC, Youden)	94.4	78.2	95.9	75	95.4
Saldir M, 2015, Turkey [20]	>72 h	50 near-term (>34 weeks) and term NICU infants: 30 infected (PS = 20%), 20 uninfected	(1) Positive blood/CSF culture or (2) negative culture, but >3 clinical signs of sepsis and abnormal laboratory results (CRP > 5 mg/dL)	Suspected sepsis, which was not supported by clinical or laboratory findings	Venous blood, 0 h (after SS)	NS	7 (ROC, NS)	93.3	95	0.96 (0.908–0.998)	96.6	90.5
Tunc T, 2015, Turkey [19]	>72 h	50 near-term (>34 weeks) and term NICU infants: 30 infected (PS = 17%), 20 uninfected	(1) Positive blood/CSF culture or (2) negative culture, but >3 clinical signs of sepsis and abnormal laboratory results (CRP > 5 mg/dL)	Suspected sepsis, which was not supported by clinical or laboratory findings	Venous blood, 0 h (after SS)	NS	7 (ROC, NS)	96.7	95	0.97 (0.918–0.998)	96.7	95
Lusyati S, 2013, Indonesia [17]	>72 h	52 preterm and term NICU infants: 18 infected (PS = 100%), 34 uninfected	Positive culture	Negative blood culture, clinically stable and no signs of infection, except mild respiratory problems treated with CPAP in the first 2 days after birth	Peripheral blood, 0 h (after SS)	Multiplex bead immunoassay	93 (ROC, NS)	72.22 (46.5–90.3)	72.22 (46.5–90.3)	NA	NA	NA
					Peripheral blood, 12 h (after SS)		25	100 (76.8–100)	80 (56.3–94.3)	NA		
					Peripheral blood, 24 h (after SS)		40	82.35 (56.6–96.2)	80 (56.3–94.3)	NA		
					Peripheral blood, 48 h (after SS)		88	64.71 (38.3–85.8)	100 (84.6–100)	NA		
		59 preterm and term NICU infants: 25 infected (PS = 0%), 34 uninfected	Negative culture, but ≥2 clinical signs of sepsis	Negative blood culture, clinically stable and no signs of infection, except mild respiratory problems treated with CPAP in the first 2 days after birth	Peripheral blood, 0 h (after SS)		28 (ROC, NS)	81.48 (61.9–93.6)	61.11 (35.8–82.6)	NA		
					Peripheral blood, 12 h (after SS)		10	70.00 (45.7–88.0)	60.00 (36.1–80.8)	NA		
					Peripheral blood, 24 h (after SS)		13	57.14 (39.4–73.7)	70.00 (45.7–88.0)	NA		
					Peripheral blood, 48 h (after SS)		3	100.00 (89.0–100.0)	31.82 (13.9–54.9)	NA		
Raynor LL, 2012, USA [2]	>72 h	226 samples from 163 preterm and term NICU infants: 128 infected (PS = 26%), 98 uninfected	(1) Positive blood culture for Gram-positive bacteria or Candida in a patient with signs of sepsis or (2) positive blood culture for Gram-negative bacteria in a patient with signs of sepsis or (3) negative blood culture but antibiotics continued ≥5 d	Negative blood culture and antibiotics for <5 d	Peripheral blood, ≤6 h (after taking the blood culture)	Multiplex antibody-coated bead array with dual-laser fluorometric detection	130 (ROC, sens = 100%)	100	28	NA	52	100
Hotoura E, 2012, Greece [16]	>72 h	82 preterm infants: 42 infected (PS = 41%), 40 healthy controls	(1) Positive blood culture and compatible signs and symptoms or (2) negative blood culture, but signs and symptoms of infection	Infection-free controls, without clinical findings or maternal risk factors for infection	Peripheral blood, 0 h (after SS), for controls at the respective days	ELISA	60 (ROC, NS)	67 (41–85)	96 (89–99)	0.95	80 (51–94)	89 (78–94)
							30	100 (78–100)	74 (63–83)	0.95	40 (30–50)	100 (90–100)
Sarafidis K, 2010, Greece [18]	>72 h	52 preterm and term NICU infants with suspected LOS: 31 infected (PS = 71%), 21 uninfected	(1) Positive blood culture (for microbes or fungi) or (2) negative blood culture, but clinical and laboratory (metabolic acidosis, thrombocytopenia, leukopenia/leukocytosis, I:T ratio ≤ 0.2 and CRP ≤ 10 mg/L) evidence of sepsis	Negative blood culture and no laboratory evidence of infection	Peripheral blood, 0 h (after SS)	ELISA	65.98 (ROC, NS)	80 (61–92)	81 (58–94)	0.892 (0.808– 0.976)	86 (67–95)	74 (59–89)
Ng PC, 2007, China [22]	>72 h	155 preterm and VLBW infants with suspected sepsis or NEC: 44 infected (PS = 59%), 111 uninfected	Confirmed episode of septicemia, meningitis, pneumonia, peritonitis, systemic fungal infection, or NEC	Episode meeting the screening criteria for suspected clinical sepsis, subsequently proven not to be infectious and improvement after antibiotic treatment was stopped between 24 and 96 h after initiation	Peripheral blood, 0 h (after SS)	Cytometric bead array (flow cytometry)	26.1 (ROC, sensitivity approaching 100% and specificity >85% or if not possible sensitivity and specificity approaching 75%)	82	82	0.88	64	92
					Peripheral blood, 24 h (after SS)		26.1	48	82	0.69	50	81
Verboon-Maciolek MA, 2006, The Netherlands [3]	NS, all infants older ≥3 days	92 preterm and term NICU infants: 66 infected (PS = 56%), 26 uninfected	(1) Positive blood culture or (2) negative blood culture but clinical sepsis	No symptoms of infection	Venous blood, 0 h (after SS)	Fully automated chemi-luminescence assay (Immulite)	60 (ROC, NS)	68 (50–82)	76 (56 –90)	NA	78 (60–91)	65 (46–80)
Arnon S, 2005, Israel [27]	NS, all infants older ≥4 days	116 preterm infants: 38 infected (PS = 61%), 78 uninfected	(1) Positive blood/CSF/urine culture (in the case of CNS 2, positive blood cultures were required) and ≥1 clinical signs of sepsis or (2) negative cultures, but ≥1 clinical signs of sepsis and 2 abnormal laboratory results persisting for >24 h	(1) Not fulfilling sepsis criteria or (2) blood taken for other reasons than infection	Peripheral blood, 0 h (after SS)	ELISA	31 (ROC, NS)	78 (65–85)	89 (79–95)	0.65 (0.35–0.76)	64 (52–76)	88 (79–95)
					Peripheral blood, 8 h (after SS)		31	47(39–51)	100 (97–100)	0.65 (0.35–0.76)	100 (93–100)	80 (68–88)
					Peripheral blood, 24 h (after SS)		31	19 (10–30)	97 (93–99)	NA	78 (67–86)	69 (57–77)
Gonzalez BE, 2003, USA [24]	>72 h	27 preterm NICU infants: 8 infected (PS = 100%), 19 uninfected	Positive blood culture	Negative blood culture	Peripheral blood, day 0 (after SS)	Quantikine kit	18 (by inspection)	75	68	NA	50	87
					Peripheral blood, day 1 (after SS)		18	75	90	NA	50	90
Ng PC, 2002, China [21]	>72 h	80 preterm and VLBW infants with 127 episodes of suspected sepsis: 32 infected (PS = 69%), 58 noninfected and 20 healthy controls	Confirmed episode of septicemia, meningitis, pneumonia, peritonitis, systemic fungal infection, or NEC	(1) Episode meeting the screening criteria for suspected clinical sepsis, subsequently proven not to be infectious or (2) healthy infant with 1–5 weeks neonatal age	Peripheral blood, 0 h (after SS)	ELISA	31 (ROC, sensitivity approaching 100% and specificity >85% or if not possible, sensitivity and specificity approaching 75%)	78	92	NA	81	91
					Peripheral blood, 12 h (after SS)		31	44	93	NA	72	81
					Peripheral blood, 24 h (after SS)		31	46	91	NA	68	80
Küster H, 1998, Germany, Slovakia, Austria [9]	>48 h	41 preterm and VLBW NICU infants: 21 infected (PS = 100%), 20 uninfected	Subjective clinical suspicion of sepsis, followed within 2 days by objective clinical evidence and sampling of specimens for positive cultures	Neither positive cultures, nor objective clinical evidence, nor subjective clinical suspicion of sepsis	Peripheral blood, day − 4 to day − 1 (diagnosis of sepsis on day 0)	ELISA	25 (ROC, maximum sens + spec)	57.1	82.9	0.94	NA	NA
					Peripheral blood, day − 4 to day 0 (diagnosis of sepsis on day 0)		25	85.7	82.9	0.94	NA	NA
					Peripheral blood, day − 4 to day + 1 (diagnosis of sepsis on day 0)		25	89.3	82.9	0.94	NA	NA
Ng PC, 1997, China [10]	>72 h	68 preterm and VLBW infants with 101 episodes of clinical suspected sepsis: 35 infected (PS = NA), 46 uninfected, 20 healthy controls	Positive blood culture or confirmed infection other than septicemia (pneumonia, peritonitis, meningitis, systemic fungal infection, and NEC) with or without positive blood culture	(1) Episode meeting the screening criteria for suspected clinical sepsis, subsequently proven not to be infectious and improvement after antibiotic treatment was stopped or (2) healthy infant with 1–8 weeks neonatal age	Peripheral blood, day 0 (after SS)	ELISA	31 (ROC, minimizing the number of misclassified episodes)	89	96	NA	95	91
					Peripheral blood, day 1 (after SS)		31	67	89	NA	84	77
Panero A, 1997, Italy [26]	>72 h	68 preterm and term NICU infants: 17 infected (PS = 82%), 51 uninfected	(1) Positive blood culture (septicemia) or (2) meningitis or (3) NEC	Uninfected controls matched for neonatal age and duration of hospital stay	Peripheral blood, 0 h (after SS)	Solid phase sandwich enzyme-amplified sensitivity immunoassay (Medgenix)	15 (NA)	100	100	NA	NA	NA

PS = Proven sepsis, NA = Not available, NS = Not specified, SS = Suspicion of sepsis, NICU = Neonatal intensive care unit, CSF = Cerebrospinal fluid, CRP = C reactive protein, AUC = Area under the curve, PPV = Positive predictive value, NPV = Negative predictive value, Sens = Sensitivity, Spec = Specificity; GA = Gestational age.

**Table 2 children-11-00486-t002:** Characteristics of IL-6 accuracy studies for the diagnosis of late onset sepsis using biomarker combinations.

Author, Year, Country, Reference	LOS Definition	Recruitment	Reference Standard in Infected Neonates	Reference Standard in Control Neonates	Sample Studied, Time of Sample Collection	Test	Biomarker Combination	Cut-Offs: IL-6 (pg/mL), sTREM-1 (pg/mL), IP-10 (pg/mL), IL-10 (pg/mL), CRP (mg/L), CD64 (Phycoerythrin-Molecules Bound Per Cell), TNF-α (pg/mL)	Sens, % (95% CI)	Spec, % (95% CI)	AUC	PPV, %	NPV, %
Dillenseger L, 2018, France [25]	>72 h	130 preterm and term NICU infants with suspected sepsis: 34 infected (PS = 53%), 96 uninfected	(1) Positive blood culture alone, or in combination with clinical signs of infection and a CRP >10 mg/L (in the case of typical skin contaminants), or meningitis (>10 cells/mL in lumbar puncture), or pneumonia (>10^4^ bacteria/mL in BAL/tracheal aspiration, positive chest radiographs, ventilator support, ≥4 clinical signs), or pyelonephritis (clinical signs of sepsis, CRP > 10 and >10^6^ cells/L and >10^5^ bacteria/mL in the urine) or (2) clinical signs and CRP ≥ 10 mg/L, no alternative diagnosis and improvement upon antibiotic treatment	(1) Clinical signs or elevated CRP explained by alternative diagnosis or positive culture, but no clinical or biological signs of infection, or positive blood culture but CRP < 4 mg/L, or antibiotic treatment <5 days or (2) clinical improvement and normalization of CRP levels without antibiotics	Peripheral blood, 0 h (after SS)	Fully automated chemiluminescence assay (Immulite)	IL-6 + CRP	IL-6: 21.7, CRP: 4.05	78.12 (60.03–90.72)	76.34 (66.40–84.54)	84.80 (75.03–96.58)	53.19 (38.08–67.89)	91.03 (82.38–96.32)
Hotoura E, 2012, Greece [16]	>72 h	82 preterm infants: 42 infected (PS = 41%), 40 healthy controls	(1) Positive blood culture and compatible signs and symptoms or (2) negative blood culture, but signs and symptoms of infection	Infection-free controls, without clinical findings or maternal risk factors for infection	Peripheral blood, 0 h (after SS), for controls at the respective days	ELISA	IL-6 + CRP	IL-6: 30, CRP: 10	100 (79–100)	96 (89–99)	NA	NA	NA
Sarafidis K, 2010, Greece [18]	>72 h	52 preterm and term NICU infants with suspected LOS: 31 infected (PS = 71%), 21 uninfected	(1) Positive blood culture (for microbes or fungi) or (2) negative blood culture, but clinical and laboratory (metabolic acidosis, thrombocytopenia, leukopenia/leukocytosis, I:T ratio ≤ 0.2 and CRP ≤ 10 mg/L) evidence of sepsis	Negative blood culture and no laboratory evidence of infection	Peripheral blood, 0 h (after SS)	ELISA	IL-6 + sTREM-1 (NS)	IL-6: 66, sTREM-1: 144	90 (73–98)	62 (38–82)	NA	77 (59-89)	81 (54–96)
Ng PC, 2007, China [22]	>72 h	155 preterm VLBW infants with suspected sepsis or NEC: 44 infected (PS = 59%), 111 uninfected	Confirmed episode of septicemia, meningitis, pneumonia, peritonitis, systemic fungal infection, or NEC	Episode meeting the screening criteria for suspected clinical sepsis, subsequently proven not to be infectious and improvement after antibiotic treatment was stopped between 24 and 96 h after initiation	Peripheral blood, 0 h (after SS)	Cytometric bead array (flow cytometry)	IL-6 + IP-10	IL-6: 26.1, IP-10: 1250 (ROC, sensitivity approaching 100% and specificity >85% or if not possible sensitivity and specificity approaching 75%)	98	72	NA	58	99
							IL-6 + IP-10 + IL-10	IL-6: 26.1, IP-10: 1250, IL-10: 7.6	98	61	NA	50	99
Verboon-Maciolek MA, 2006, The Netherlands [3]	NS, all infants ≥ 3 days	92 preterm and term NICU infants: 66 infected (PS = 56%), 26 uninfected	(1) Positive blood culture or (2) negative blood culture but clinical sepsis	No symptoms of infection	Venous blood, 0 h (after SS)	IL-6: fully automated chemiluminescence assay (Immulite), CRP: rate nephelometry	IL-6 + CRP	IL-6: 60, CRP: 14	92 (78–98)	41 (24–61)	NA	67 (54–80)	80 (52–96)
Ng PC, 2002, China [21]	>72 h	80 preterm VLBW infants with 127 episodes of suspected sepsis: 32 infected (PS = 69%), 58 noninfected and 20 healthy controls	Confirmed episode of septicemia, meningitis, pneumonia, peritonitis, systemic fungal infection, or NEC (stage II or above in Bell’s classification)	(1) Episode meeting the screening criteria for suspected clinical sepsis, subsequently proven not to be infectious or (2) healthy infant with 1–5 weeks neonatal age	Peripheral blood, 0 h (IL-6) and 24 h (CD64) after SS	IL-6: ELISA, CD64: flow cytometry	IL-6 + CD64	IL-6: 31, CD64: 4000 (ROC, sensitivity approaching 100% and specificity >85% or if not possible sensitivity and specificity approaching 75%)	100	86	NA	74	100
					Peripheral blood, 24 h (after SS)		IL-6 + CD64		97	86	NA	73	99
					peripheral blood, 48 h (IL-6) and 24 h (CD64) after SS		IL-6 + CD64		95	83	NA	70	97
Ng PC, 1997, China [10]	>72 h	68 preterm VLBW infants with 101 episodes of clinical suspected sepsis: 35 infected (PS = NA), 46 uninfected, 20 healthy controls	Positive blood culture or confirmed infection other than septicemia (pneumonia, peritonitis, meningitis, systemic fungal infection, and NEC) with or without positive blood culture	(1) Episode meeting the screening criteria for suspected clinical sepsis, subsequently proven not to be infectious and improvement after antibiotic treatment was stopped or (2) healthy infant with 1–8 weeks neonatal age	Peripheral blood, day 0 (after SS)	IL-6+TNF-α: ELISA, CRP: turbidity assay	IL-6 + CRP	IL-6: 31, CRP: 12 (ROC, sensitivity approaching 100% and specificity >85% or if not possible sensitivity and specificity approaching 75%)	93	96	NA	95	95
					Peripheral blood, day 1 (after SS)		IL-6 + CRP		93	88	NA	86	94
					Peripheral blood, day 0 (after SS)		IL-6 + TNF-α		95	84	NA	83	96
					Peripheral blood, day 1 (after SS)		IL-6 + TNF-α		91	84	NA	82	92
					Peripheral blood, day 0 (after SS)		IL-6 + CRP + TNF-α		95	84	NA	82	96
					Peripheral blood, day 1 (after SS)		IL-6 + CRP + TNF-α		98	80	NA	80	98
					Peripheral blood, day 0 (IL-6+CRP) and day 1 (TNF-α) after SS		IL-6 + CRP + TNF-α		98	91	NA	90	98
					Peripheral blood, day 0 (IL-6+CRP) and day 2 (CRP) after SS		IL-6 + CRP		98	91	NA	90	98

PS = Proven sepsis, NA = Not available, NS = Not specified, SS = Suspicion of sepsis, NICU = Neonatal intensive care unit, CRP = C reactive protein, PC = Platelet count, AUC = Area under the curve, PPV = Positive predictive value, NPV = Negative predictive value, Sens = Sensitivity, Spec = Specificity.

**Table 3 children-11-00486-t003:** Subgroup analysis of IL-6 accuracy studies for diagnosis of late onset sepsis.

Subgroup		No. Studies	Pooled Sensitivity, %	Pooled Specificity, %
Study population	Preterm	8	86.59	85.71
	Preterm and term	6	81.77	86.05
Timing	0 h *	11	84.22	85.83
	≤12 h *	3	56.82	93.68
	≤24 h *	6	54.29	88.34
	≤48 h *	3	67.21	92.44
Sepsis definition	Culture proven only	3	84.62	74.36
Study design	Blinding	2	80.77	80.00
Biomarker combinations	IL-6 + CRP	4	92.09	78.95

* Time after suspicion of sepsis.

**Table 4 children-11-00486-t004:** Quality of IL-6 diagnostic accuracy studies for diagnosis of late onset sepsis from 1990 to 2020 according to the STARD criteria (“Standards of Reporting Diagnostic Accuracy Studies” [12]).

Quality of Reporting of IL-6 Accuracy Studies for Diagnosing Late (>72 h) Onset Infection		
Category and Item No.	YES	NO
Methods: participants		
Describe the study population:		
1A. The inclusion and exclusion criteria	10	6
1B. Setting, and locations where data were collected	15	1
Describe participant recruitment:		
2A. Was enrollment of patients based only on clinical signs suggesting infection?	12	4
2B. Were such patients consecutively enrolled?	2	10
2C. Was enrollment of patients based only on maternal risk factors for infection?	0	16
2D. Were such patients consecutively enrolled?	0	0
2E. Were patients identified by searching hospital records?	0	16
2F. Did the study include both patients already diagnosed with sepsis and participants in whom sepsis had been excluded?	2	14
Describe data collection:		
3. Was data collection planned before the index test and reference standard were performed (prospective study)?	14	2
Test methods		
Methods pertaining to the reference standard and the index test:		
4A. Was a composite reference standard used to identify all newborns with sepsis, and verify index test results in infected babies?	13	3
4B. Was a reference standard used to exclude sepsis?	14	2
4C. Was a composite reference standard used to identify all newborns without sepsis, and verify index test results in uninfected babies?	4	10
4D. Did the index test or its comparator form part of the reference standard?	2	14
5. Were categories of results of the index test (including cut-offs) and the reference standard defined after obtaining results?	16	0
6. Did the study report the number, training and expertise of the persons executing and reading the index tests and the reference standard?	3	13
7. Was there blinding to results of the index test and the reference standard?	4	12
Statistical methods		
8. Describe the statistical methods used to quantify uncertainty (i.e., 95% confidence intervals)	6	10
9. Describe methods for calculating test reproducibility	4	12
Results: participants and test results		
10A. Describe when the study was carried out, including beginning and ending dates of recruitment	13	3
10B. Did the study report clinical and demographic (postnatal hours or days, gestational age, birth weight, gender) features in those with and without sepsis?	15	1
10C. Did the study report distribution of illness severity scores in those with and without sepsis?	0	16
11. Report the number of participants satisfying the criteria for inclusion that did or did not undergo the index tests and/or or the reference standard; describe why participants failed to receive either test	4	12
12. Report a cross-tabulation of the results (including indeterminate and missing results) using the results of the reference standard; for continuous results, report the distribution of the test results using the results of the reference standard	2	14
Results: estimates		
13. Report measures of statistical uncertainty (i.e., 95% confidence intervals)	6	10
14. Report how indeterminate results, missing responses and outliers of index tests were handled	1	15
15. Report estimates of test reproducibility	5	11

## Data Availability

Not applicable, as all data are published studies.

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
