# Peer review of "Reliability of IL-6 Alone and in Combination for Diagnosis of Late Onset Sepsis: A Systematic Review"

_children, 2024, doi:10.3390/children11040486_

Round 1

Reviewer 1 Report

Comments and Suggestions for Authors

The study focuses on evaluating the diagnostic accuracy of Interleukin-6 (IL-6) as a marker for neonatal sepsis, specifically late-onset sepsis (LOS), in both term and preterm infants after 72 hours of life. A comprehensive search of IL-6 diagnostic accuracy studies from 1990 to 2020 was conducted using the PubMed database, followed by a accurate study selection process.

Some major concerns

1. Regarding material and methods section, a more detailed breakdown of the search terms and criteria, as well as their rationale, would enhance transparency and reproducibility.

2. it would be beneficial to elaborate on how discrepancies or disagreements during data extraction were resolved to ensure the reliability of the extracted information.

3. The use of the adapted STARD checklist is appropriate for assessing the quality of studies. However, it would be helpful to mention any predefined criteria for considering a study as high-quality or low-quality and how the checklist items were weighted in the analysis.

4. Providing details on the predefined thresholds or criteria for subgroup analysis would add clarity.

5. Include the reasons for exclusion at each stage, especially for the 23 full-text articles that were assessed but not included in the meta-analysis.

6. Additionally, providing a rationale for the choice of specific biomarker combinations or discussing potential clinical implications would add value.

7. Providing more details on how reproducibility and statistical analyses were conducted in the studies reporting this information would enhance the transparency of the research process.

8. The manuscript outline is not clearly outlined for publication in the journal.

Author Response

Reply to REVIEWER 1

The study focuses on evaluating the diagnostic accuracy of Interleukin-6 (IL-6) as a marker for neonatal sepsis, specifically late-onset sepsis (LOS), in both term and preterm infants after 72 hours of life. A comprehensive search of IL-6 diagnostic accuracy studies from 1990 to 2020 was conducted using the PubMed database, followed by a accurate study selection process.

Some major concerns

  1. Regarding material and methods section, a more detailed breakdown of the search terms and criteria, as well as their rationale, would enhance transparency and reproducibility.

REPLY:

Studies eligible for review inclusion were retrieved using the PubMed database including diagnostic accuracy studies of IL-6 in neonates published between 1990 and 2020. The combined search term used was: (Interleukin-6 OR IL-6) AND (neonatal sepsis OR neonatal infection OR sepsis) AND (late onset sepsis OR LOS OR LONS). No PubMed filters or language restrictions were used.

Using these search terms everyone finds the same literature. We chose to use no language restriction since the authors mother tongue is German, so publications in German could have been included. However, all publications included were in English.

  1. it would be beneficial to elaborate on how discrepancies or disagreements during data extraction were resolved to ensure the reliability of the extracted information.

REPLY: We added: Two investigators (JE, ER) independently performed the data extraction of all included studies. In case of discrepancies these were resolved by the senior author (BR).

  1. The use of the adapted STARD checklist is appropriate for assessing the quality of studies. However, it would be helpful to mention any predefined criteria for considering a study as high-quality or low-quality and how the checklist items were weighted in the analysis.

REPLY: We performed additionally the quality assessment of diagnostic accuracy studies (QUADAS) tool including 11 questions. Questions with “yes”, “no”, and “unknown” answer, were scored as 1, -1, and 0, respectively [Whiting et al]. Thus, we could confirm our first analysis using the STARD criteria.

We added the following reference:

Whiting P, Rutjes AW, Reitsma JB, Bossuyt PM, Kleijnen J. The development of QUADAS: a tool for the quality assessment of studies of diagnostic accuracy included in systematic reviews. BMC Med Res Methodol. 2003 Nov 10; 3:25

  1. Providing details on the predefined thresholds or criteria for subgroup analysis would add clarity

REPLY: Our subgroup analyses included the following: gestational age (preterm vs. term infants and mixed populations), time of sample collection, and combined markers (e.g.IL-6 and CRP) provided there were at least three or more studies.

  1. Include the reasons for exclusion at each stage, especially for the 23 full-text articles that were assessed but not included in the meta-analysis.

REPLY: Study selection was performed according to PRISMA criteria and summarized in the flow chart in Figure 1. In the PRISMA criteria including the reasons for exclusion is intended only at the level of assessing full texts. At this level we excluded 5 studies, 4 because they did not distinguish between EONS and LOS, one because it wasn’t a diagnostic accuracy study, this information is found in the flow chart. Hence, 18 Studies were included in the qualitative analysis, two of these studies did not provide statistical measures like Sensitivity and Specificity and therefore did not form part of the meta-analysis. Reasons for exclusion at the abstract level are already described in the following paragraph in the methods section:”The following criteria had to be fulfilled by reviewing the abstract: only neonates presenting with culture proven and/or clinically suspected sepsis and IL-6 (alone or combined with other inflammatory markers) being evaluated regarding its potential for the diagnosis of LOS. We excluded all studies dealing with early-onset sepsis (EONS) or other neonatal bacterial infections, all studies written in other languages than English or German, animal and in vitro studies.”Even though we recorded the reasons for exclusion for each of the 24 papers excluded at abstract level, we don’t think the paper would benefit from a more detailed list.

  1. Additionally, providing a rationale for the choice of specific biomarker combinations or discussing potential clinical implications would add value.

REPLY: We touched on the topic in the introduction where we wrote that the idea is to combine the early and sensitive IL-6 with later and more specific markers like CRP. In the methods section (last sentence) we defined, that biomarker combinations would be assessed in the subgroup analysis, if at least three of our included studies analysed that exact biomarker combination. Unfortunately this was only the case for the combination of IL-6 and CRP. The overall small number of studies analysing biomarker combinations was mentioned as limitation of our study in the corresponding section. Hence, we are not able to deduct clinical implications from our findings. We did however discuss the findings of the individual studies in the discussion section (3rd paragraph) as well as the clinical applicability (last paragraph of the discussion section, and answer to reviewer 3)

  1. Providing more details on how reproducibility and statistical analyses were conducted in the studies reporting this information would enhance the transparency of the research process.

REPLY: We clearly determined main analyses and subgroup analyses. Forest plots show all included studies assessed and sensitivity and specificity analysis was done appropriate. Hence, all main tools can be evaluated by any interested reader, hence, we can´t find difficulties in reproducibility of the data. Additionally, all studies included are presented in detail (see tables). Only 6 and 4 studies, respectively provided information on their statistical methods and their methods for calculating test reproducibility as seen in table.4.

  1. The manuscript outline is not clearly outlined for publication in the journal.

REPLY: We adapted the format accordingly.

Reviewer 2 Report

Comments and Suggestions for Authors

The authors present a meta-analysis of 16 studies evaluating the usefulness of IL-6 in the diagnosis of late sepsis in neonates. The problem with this molecule, although it is very sensitive and specific, and rises earlier than CRP or procalcitonin, is laboratory availability. Some centres do not have a laboratory for this test, and most importantly, it is not usually available on an urgent basis, but only in the morning scheduled blood test, so the clinical usefulness of this molecule is limited. This should be added to the Discussion of the manuscript. The tables and figures are adequately presented, both forest plots. I miss a funnel plot showing the publication bias study of the meta-analysis.

It is interesting to add a forest plot of the sensitivity and specificity of IL-6 in neonates, including only the 8 articles analysing this data.

The authors should rewrite part of the manuscript with other words, as the percent match is 33% according to the iThenticate report, mainly in the introduction.

In addition, throughout the manuscript the text has different fonts and sizes, so this also needs to be corrected.

Comments on the Quality of English Language

Moderate editing of English language required. The authors should rewrite part of the manuscript with other words, as the percent match is 33% according to the iThenticate report, mainly in the introduction.

Author Response

Reply to REVIEWER 2

  • The authors present a meta-analysis of 16 studies evaluating the usefulness of IL-6 in the diagnosis of late sepsis in neonates. The problem with this molecule, although it is very sensitive and specific, and rises earlier than CRP or procalcitonin, is laboratory availability. Some centres do not have a laboratory for this test, and most importantly, it is not usually available on an urgent basis, but only in the morning scheduled blood test, so the clinical usefulness of this molecule is limited. This should be added to the Discussion of the manuscript. The tables and figures are adequately presented, both forest plots. I miss a funnel plot showing the publication bias study of the meta-analysis.

REPLY: Your criticism is understandable, but this systematic review did neither intend to address the problem of clinical availability of IL-6 nor solve that anyway. At a tertiary level hospital like ours IL-6 is available 24 hours 7 days.

We described bias problems in detail in the results section and, thus, did not feel that a funnel plot would add further information

  • It is interesting to add a forest plot of the sensitivity and specificity of IL-6 in neonates, including only the 8 articles analysing this data.

REPLY: From the question we are not sure which 8 articles are meant. Maybe the 8 articles that had an only preterm study population. While we agree that additional forest plots could give a sense of heterogeneity within each subgroup, with a total of 15 studies these eight studies and other subgroups are easily identified in the overall forest plot.

  • The authors should rewrite part of the manuscript with other words, as the percent match is 33% according to the iThenticate report, mainly in the introduction.

REPLY: Was done accordingly (actual body of the manuscript and abstract 24%)

  • In addition, throughout the manuscript the text has different fonts and sizes, so this also needs to be corrected.

REPLY: We made sure fonts and sizes were uniform throughout the manuscript

  • Moderate editing of English language required. The authors should rewrite part of the manuscript with other words, as the percent match is 33% according to the iThenticate report, mainly in the introduction.

REPLY: Was done accordingly.

Reviewer 3 Report

Comments and Suggestions for Authors

Eichberger and colleagues present a systematic review of the performance characteristics of serum IL-6 levels as a test for the diagnosis of late-onset neonatal sepsis. The authors conclude that the test has good sensitivity and reasonable sensitivity, despite the heterogeneity and small number of studies available for analysis. The authors used a modified STARD checklist to assess the quality of the studies, yet the fundamental problem of the variable and “soft” criteria used for diagnosing sepsis does not receive sufficient attention, nor is it emphasized in the Discussion. Details follow.

The most significant issue regards the definition of sepsis, in the Methods. The authors discuss heterogeneity in the definition of late onset sepsis with respect to the timing of onset being beyond 72 hours as opposed to beyond 48 hours, but this variation is of little epidemiological and practical significance.  Much more important, and critically so, are the criteria used to diagnose late onset sepsis, irrespective of the exact time of its onset.  The variability in these definitions is important, and it may contribute to the discrepant rates of treatment for culture-negative sepsis across different NICUs; this may also be the primary factor which defines the performance characteristics of the test, assuming that the IL-6 assays are not wildly discrepant.  Consequently, the authors need to be clear about what each study defines as an "infected neonate", both in the language used and in detail provided.  For example, in table 1, page 9, first row, fourth column, the definition reported is "positive blood culture in addition to clinical signs and abnormal acute phase reactants".  Using standard logical operators, this language may be construed to indicate that neonates classified as infected have a positive blood culture AND clinical signs AND abnormal acute phase reactants; however, this is unlikely, so the study may have classified as infected those neonates with positive blood culture OR clinical signs OR abnormal acute phase reactants. This is probably also incorrect: considering the clinical context and this study, the most likely correct expression of the definition used is positive blood culture OR (clinical signs AND abnormal acute phase reactants).  These considerations may not be dismissed as semantics, and the authors should carefully go through the wording and the reference standards for both infected and control neonates and to clarify the criteria used in the various studies.  It would be useful to include a column for rates of late onset sepsis in the individual unit, as defined in each study – or, a note about this information being unavailable (which I imagine is the case in most instances).

A more useful and likely more available variable about the nature and quality of each study would be the proportion of all babies classified as infected who had positive blood and/or CSF cultures; a study of late-onset sepsis in NICU in which 80% of cases are culture-positive is very different from one in which only 8% of cases are culture positive. In some studies, this may not be available.  I believe this quality measure would relate to the risk of bias in the study.  Since the biomarker results are typically not blinded, and in some cases, even more problematically, the biomarker test results are used to classify babies as having infection present or absent, this generates a substantial bias that artificially inflates the accuracy of the tests, both with respect to their sensitivity and specificity (for example, if elevated IL-6 levels constitute a sufficient criterion to define infection, and the results are available to the clinicians, then every baby with a “negative” IL-6 will be negative for sepsis, i.e., we have artificially inflated the specificity). When evaluating the relationship of a test result to a disease, researchers must ensure that the disease was classified independently of the test results.

The Discussion then needs to address in much more detail this important issue.

The inclusion of studies such as ref. #19, which conflate pneumonia and NEC with septicemia/culture-proven sepsis, also contributes to confusion about the source data, and diminish the reliability of any conclusions the authors may wish to derive.  Indeed, if the authors perform a rigorous evaluation of existing studies and find that none provide reliable data on  the usefulness of IL-6 in late onset neonatal sepsis, that would be a valuable conclusions, leading to a recommendation that such (reliable) studies need to be performed.

In page 8 of the Discussion, the authors state that "antibiotics could be confidently discontinued at 48 hours without waiting for microbiological results…".  However, negative results of blood cultures at 48 hours (and even at 36 hours) are virtually final, and in that case the decision to discontinue antibiotics at 48 hours in infants who are in good clinical condition does not need to rely on any biomarker testing.

The Conclusion paragraph on page 8 is a summarized restatement of findings, not a true conclusion (i.e., authors’ inference from the findings).

Minor issues:

Please consider making the following corrections:

Page 4, … Figure 1 “depicts”.

Page 5, … Measures of statistical uncertainty “were” reported….

Page 6, last paragraph, … PPV between patients with[out]? sepsis and those with sepsis….

Page 6, 3 lines from the bottom, serial measurements of inflammatory markers "can further" improve diagnostic accuracy.

Comments on the Quality of English Language

minor English language edits suggested

Author Response

Reply to REVIEWER 3

Eichberger and colleagues present a systematic review of the performance characteristics of serum IL-6 levels as a test for the diagnosis of late-onset neonatal sepsis. The authors conclude that the test has good sensitivity and reasonable sensitivity, despite the heterogeneity and small number of studies available for analysis. The authors used a modified STARD checklist to assess the quality of the studies, yet the fundamental problem of the variable and “soft” criteria used for diagnosing sepsis does not receive sufficient attention, nor is it emphasized in the Discussion. Details follow. 

The most significant issue regards the definition of sepsis, in the Methods. The authors discuss heterogeneity in the definition of late onset sepsis with respect to the timing of onset being beyond 72 hours as opposed to beyond 48 hours, but this variation is of little epidemiological and practical significance.  Much more important, and critically so, are the criteria used to diagnose late onset sepsis, irrespective of the exact time of its onset.  The variability in these definitions is important, and it may contribute to the discrepant rates of treatment for culture-negative sepsis across different NICUs; this may also be the primary factor which defines the performance characteristics of the test, assuming that the IL-6 assays are not wildly discrepant.  

REPLY: We addressed these concerns in the answers below and added a section, summarizing our answers, to the Discussion.

Consequently, the authors need to be clear about what each study defines as an "infected neonate", both in the language used and in detail provided.  For example, in table 1, page 9, first row, fourth column, the definition reported is "positive blood culture in addition to clinical signs and abnormal acute phase reactants".  Using standard logical operators, this language may be construed to indicate that neonates classified as infected have a positive blood culture AND clinical signs AND abnormal acute phase reactants; however, this is unlikely, so the study may have classified as infected those neonates with positive blood culture OR clinical signs OR abnormal acute phase reactants. This is probably also incorrect: considering the clinical context and this study, the most likely correct expression of the definition used is positive blood culture OR (clinical signs AND abnormal acute phase reactants).  These considerations may not be dismissed as semantics, and the authors should carefully go through the wording and the reference standards for both infected and control neonates and to clarify the criteria used in the various studies.  

REPLY: We are aware of the need for correct usage of logical operators in the definitions of the reference standard. For the mentioned study (ref #21) the definition of infected neonates was actually exactly as written in our table, as this study relied on culture positive infants and healthy controls. See original text from the paper below. Infants who were born ≤32 weeks gestational age and 4to 60 days postnatal age with gram-positive and/or negative bacteria detected in blood culture were included in the study. Patients were diagnosed with ‘culture-proven sepsis’ when the blood culture was positive in addition to clinical signs and abnormal acute phase reactants.

Patients were matched in terms of gestational age, birth weight and gender. Newborn infants who had no signs or symptoms of sepsis were enrolled in the control group. Blood cultures were not taken from control group patients. A total of 55 patients were included in the study. Of these, 26 had culture-proven LOS. Twenty-nine patients represented the control group.

We have addressed this also in the text of our work: In two studies, the sepsis group consisted only of culture-proven cases (21, 22).

One might argue that the positive blood culture was the most stringent criteria. Nevertheless the presence of clinical signs and abnormal acute phase reactants is also correct, since the authors also had a definition for a probable sepsis group consisting of infants with clinical signs and abnormal acute phase reactants but negative blood culture (see below). However, this group did not form part of the analysis

REPLY: The presence of at least two clinical findings and at least two laboratory signs with accompanying negative blood culture was defined as ‘probable sepsis’.

We made sure to check once more, whether the definitions in our table were correct and the wording was clear.

R3: It would be useful to include a column for rates of late onset sepsis in the individual unit, as defined in each study – or, a note about this information being unavailable (which I imagine is the case in most instances).

REPLY: See point below!

R3: A more useful and likely more available variable about the nature and quality of each study would be the proportion of all babies classified as infected who had positive blood and/or CSF cultures; a study of late-onset sepsis in NICU in which 80% of cases are culture-positive is very different from one in which only 8% of cases are culture positive. In some studies, this may not be available.  I believe this quality measure would relate to the risk of bias in the study.  

REPLY: We agree that this information might be of interest to the reader and included this data. We further added the following section (see below).

R3: While mixed sepsis groups introduce differential verification bias they are closer to the clinic where clinically suspected sepsis in a sick infant cannot be discarded solely by a negative blood culture. The percentage of culture proven sepsis (=PS) in the group of infected infants can be found in table 1 and 2 under recruitment.

R3: Since the biomarker results are typically not blinded, and in some cases, even more problematically, the biomarker test results are used to classify babies as having infection present or absent, this generates a substantial bias that artificially inflates the accuracy of the tests, both with respect to their sensitivity and specificity (for example, if elevated IL-6 levels constitute a sufficient criterion to define infection, and the results are available to the clinicians, then every baby with a “negative” IL-6 will be negative for sepsis, i.e., we have artificially inflated the specificity). When evaluating the relationship of a test result to a disease, researchers must ensure that the disease was classified independently of the test results. The Discussion then needs to address in much more detail this important issue.

REPLY: The absence of appropriate blinding is an issue in these studies especially with post hoc definition of cut-off values. The STARD criteria checks whether information on blinding was reported in the individual study, which for our meta-analysis was the case in four studies as described.

To our knowledge based on the information provided by the individual studies, none of the included studies had elevated IL-6 levels in their reference criteria to identify septic infants. We highly doubt that a reference standard including the index test of the study would have gone unnoticed by the reviewers of that study. There is an issue regarding the inclusion of CRP in the reference standard as it often forms the comparator of the index test, and by nature of the inflammatory cascade is related to IL-6, which we have addressed here:

“As an inflammatory marker, CRP serves as an important comparator of the index test, however in two studies it also formed part of the reference standard for sepsis diagnosis [21, 23].” We further added: “Shortly, incorporation bias occurs if the index test or the comparator of the index test form part of the reference standard. This fact distorts the diagnostic abilities of any marker when used as a comparator of the index test, but also combinations of biomarkers including the one or markers related to the one parameter are biased [28].”.

The inclusion of studies such as ref. #19, which conflate pneumonia and NEC with septicemia/culture-proven sepsis, also contributes to confusion about the source data, and diminish the reliability of any conclusions the authors may wish to derive.  Indeed, if the authors perform a rigorous evaluation of existing studies and find that none provide reliable data on the usefulness of IL-6 in late onset neonatal sepsis, that would be a valuable conclusions, leading to a recommendation that such (reliable) studies need to be performed.

REPLY: The issue of including pneumonia and NEC in the criteria for the septic infant was present in 4 studies in our paper, three studies by Ng et. al (ref #10, #18 and #19) and one by Panero et al. (ref # 23). The inclusion of these papers into the meta-analysis is justified by the fact that in the recruitment process (3rd column) all children (consecutively separated into infected and control neonates) had to meet screening criteria for suspected clinical sepsis (ref #10, #18 and #19), which was a sufficient criterion in other studies included (which did not make further distinctions regarding the focus of infection). In the study by Panero et al. (ref #23) this was not the case. However, if looking at the exact numbers we find that, of the 17 infants, 12 had a positive blood culture, 2 had meningitis and only 3 had NEC (with a negative systemic culture result), which would actually correspond to a high positive culture rate of 82% in the group of infants classified as infected in this study (see comment # of reviewer 3). Furthermore, another recent metaanalysis by Sun et al. (2019) also included two of the mentioned papers (ref #10 and #19) and supports our decision to include these studies in our review.

R3: In page 8 of the Discussion, the authors state that "antibiotics could be confidently discontinued at 48 hours without waiting for microbiological results…".  However, negative results of blood cultures at 48 hours (and even at 36 hours) are virtually final, and in that case the decision to discontinue antibiotics at 48 hours in infants who are in good clinical condition does not need to rely on any biomarker testing.

REPLY: We rewrote that section to:

While withholding of antibiotic treatment at the onset of sepsis is not recommended, high sensitivity (98%) and negative predictive values (98%) of this combination indicate that antibiotics could be discontinued at 48 hours, provided that the infants were in good clinical condition (10). This finding can only complement the already common practice of empirically treating the infant for at least 48 hours awaiting blood culture results. In the era of continuously monitored blood culture systems several studies have even challenged this time frame. A study investigating the time-to-positivity (TTP) of blood cultures in children with proven sepsis, found that 90% of blood cultures were positive within 36 hours, in most cases even <24 hours, of incubation. They concluded that discontinuing empirical treatment in the absence of a positive blood culture should already be considered after 24 and 36 h (citation below).

Dierig A, Berger C, Agyeman PKA, Bernhard-Stirnemann S, Giannoni E, Stocker M, Posfay-Barbe KM, Niederer-Loher A, Kahlert CR, Donas A, Hasters P, Relly C, Riedel T, Aebi C, Schlapbach LJ, Heininger U; Swiss Pediatric Sepsis Study. Time-to-Positivity of Blood Cultures in Children With Sepsis. Front Pediatr. 2018 Aug 8;6:222. doi: 10.3389/fped.2018.00222. PMID: 30135859; PMCID: PMC6092514.

The Conclusion paragraph on page 8 is a summarized restatement of findings, not a true conclusion (i.e., authors’ inference from the findings).

REPLY: In our opinion the findings of the subgroup-analysis do constitute the conclusion of this meta-analysis and belong in the conclusion section. We added a few sentences on our inference from the findings.

Based on the findings of this review IL-6 might be of use for the diagnosis of late onset sepsis in the population of preterm infants when measured at the time of sepsis suspicion. Evaluation of these results in the context of existing literature is difficult since other reviews on this topic include either mixed study populations or even sepsis groups consisting of early and late onset sepsis cases. To confirm the use of IL-6 in the diagnosis of LOS more prospective studies in well-defined study populations and with well-defined sepsis criteria are needed.

Minor issues: Please consider making the following corrections:
Page 4, … Figure 1 “depicts”.
Page 5, … Measures of statistical uncertainty “were” reported….
Page 6, last paragraph, … PPV between patients with[out]? sepsis and those with sepsis….
Page 6, 3 lines from the bottom, serial measurements of inflammatory markers "can further" improve diagnostic accuracy.

REPLY: Corrections were made accordingly

Round 2

Reviewer 1 Report

Comments and Suggestions for Authors

Authors have updated the reviewers comments in the revised version.

But still the version is not significant and give necessary information.

Author Response

Thank you for accepting the revised manuscript.

Reviewer 2 Report

Comments and Suggestions for Authors

The authors have adequately answered the questions raised by the reviewers.

Author Response

(The authors gave the same response as above.)

Reviewer 3 Report

Comments and Suggestions for Authors

The authors made a substantial effort to revise the manuscript, and to Reply to the issues this reviewer pointed out. However, there are some changes they did NOT make, despite indicating in their reply that they did. Also, other "responses" seem to be deferred to a later response but then it is not addressed directly (specifics follow). 

Is it possible that the authors did not upload the latest version with corrections? I looked at the latest "revised" version in PDF (which has highlighted changes but doesn't have line numbers, and the tables are very difficult to read), and the Word version with Track Changes ("children-2903682-supplementary.docx").  

Briefly, the authors have satisfactorily addressed most comments. A few issues remain. 

1. I had suggested that "It would be useful to include a column with rates of late-onset sepsis..." in each individual study NICU. The authors replied "See point below", and later stated that they "...agree ... and included this data...". However, I could not find the extra column in the tables, nor do I see those data elsewhere in the paper. I am not sure if the authors understood the point. The question is, what was the overall rate (as % of admissions) of late onset sepsis in each NICU from which study data were derived? Also, how did each NICU define such rate? Positive blood/CSF cultures only, all treated (culture positive or negative) sepsis, or both rates reported separately? I understand that the authors report numbers of "infected" and "uninfected" infants (which is confusing terminology), but the question was about overall rates in the context of each NICU. I also noted that many studies might not have provided such information, which would be useful to note.

2. Under "minor issues", I asked the authors to make some corrections, , to which they replied: "Corrections were made accordingly". However, NONE of the corrections seem to have been made.

For example, under Results, it still reads: "Figure 1 depictures...", instead of "depicts..."

I will not retype all the other corrections or errors. Please see that section of the original comments from Reviewer 3 and make all the suggested corrections.

In addition, in line 506 of the latest version (Word document), reference "221" is in error.

Finally, for the new sentence beginning with "Shortly," I suggest changing this to "Briefly," or, "In short," in more standard English.

Comments on the Quality of English Language

Minor English issues which the authors could correct quickly, but they have not

Author Response

Reply to reviewer 3, second revision:

We are sorry about the inconvenience. The corresponding author thought the minor corrections were already done by the colleagues, but this was a misunderstanding.

  1. Rates of LOS in each individual study NICU:

The reviewer asked for rates of LOS from each study included in the analysis. However, as already suspected by the reviewer himself these data were not provided. Therefore, we included the percentage of culture proven sepsis cases within the sepsis group of each study. We added in table 1 and 2 under the column “Recruitment” the percentage of culture proven sepsis (PS = proven sepsis ….%).

To better read the tables we changed the page orientation to landscape format.

  1. Minor issues: Please consider making the following corrections:

Page 4, … Figure 1 “depicts”. We corrected depictures to depicts.

Page 5, … Measures of statistical uncertainty “were” reported…. We corrected “was” to “were”

Page 6, last paragraph, … PPV between patients with[out]? sepsis and those with sepsis…. We corrected to “….without sepsis and those with sepsis…”

Page 6, 3 lines from the bottom, serial measurements of inflammatory markers "can further" improve diagnostic accuracy. As suggested we wrote “…. can further improve diagnostic accuracy”

Line 556, reference 221?: was corrected to … [21,22]

Page 8: Shortly, incorporation bias … was corrected to: In brief, incorporation bias …